# Change, Adversity, and Adaptation: Young People’s Experience of the COVID-19 Pandemic Expressed through Artwork and Semi-Structured Interviews

**DOI:** 10.3390/ijerph21050636

**Published:** 2024-05-16

**Authors:** Rhiannon Thompson, Lucy Brown, Rakhi Biswas Evans, Rayhan Mahbub, Amelia Rees, Molly Wilson, Lindsay H. Dewa, Helen Ward, Mireille B. Toledano

**Affiliations:** 1School of Public Health, Imperial College London, London W12 0BZ, UK; r.thompson19@imperial.ac.uk (R.T.); h.ward@imperial.ac.uk (H.W.); 2National Institute for Health and Care Research (NIHR), School for Public Health Research (SPHR), London, UK; 3Medical Research Centre (MRC)—Centre for Environment and Health, School of Public Health, Imperial College London, London W12 0BZ, UK; 4Institute of Zoology, Zoological Society, London NW1 4RY, UK; 5Public Contributor, Imperial College, London, UK; 6Independent Consultant, London, UK; molly.wilson@cssd.ac.uk; 7The Royal Central School of Speech and Drama, London NW3 3HY, UK; 8NIHR Imperial Biomedical Research Centre (BRC), Imperial College London, London W2 1NY, UK; 9NIHR Applied Research Collaboration North West London, London, UK; 10Imperial College Healthcare NHS Trust, London, UK; 11Mohn Centre for Children’s Health and Wellbeing, School of Public Health, Imperial College London, London W12 0BZ, UK; 12NIHR Health Protection Research Unit (HPRU) in Environmental Exposures and Health, School of Public Health, Imperial College London, London W12 0BZ, UK; 13NIHR HPRU in Chemical and Radiation Threats and Hazards, School of Public Health, Imperial College London, London W12 0BZ, UK

**Keywords:** COVID-19, lockdown, mental health, adolescence, qualitative, co-production

## Abstract

This study explores how young people’s mental health was affected by the COVID-19 pandemic using artwork and semi-structured interviews. The mental health impacts of the pandemic are important to understand so that policy and practice professionals can support those affected, prepare and respond to future crises, and support young people who are isolated and restricted in other contexts. Co-designed participatory art workshops and interviews were conducted with 16–18-year-olds (*n* = 21, 62% female) from the London-based Longitudinal cohort Study of Cognition, Adolescents and Mobile Phones (SCAMP). Artworks and interview transcripts were qualitatively co-and analysed with young people. From interviews, six themes were identified: adaptation, restriction, change, challenges, overcoming adversity, and lockdown life. From the artwork, four themes were identified: trapped, negative mental wellbeing, positive emotions, and technology. Everyday factors such as home environment, social support, hobbies, habits, and online education were key determinants of how challenged and restricted participants felt, and their capacity to overcome this. This demonstrates the importance of wider (social and environmental) determinants and supports a systems-level public health approach to young people’s mental health. For example, young people’s mental health services should collaborate with other sectors to address such determinants in a holistic way. Clearer guidance and support with occupation, relationships, environment, routine and activities could mitigate the negative mental health impacts of major environmental changes on young people.

## 1. Introduction

First identified in 2019, coronavirus (COVID-19) was officially declared a pandemic by the World Health Organisation (WHO) in March 2020, and it is estimated that it was associated with 14.9 million excess direct and indirect deaths worldwide across 2020 and 2021 [1]. In the UK, a series of government-enforced behavioural interventions termed “lockdowns” were introduced as a public health measure to reduce the spread of the virus; all non-essential public places were closed, and people were told to stay at home. For young people, this meant substantial disruption to their education and daily activities, preventing them from studying, socialising, working, and living as usual for over a year.

The pandemic and lockdowns raised concerns about young people’s mental health. Although mental health difficulties and demands on services were already increasing in the UK pre-pandemic [2], increased anxiety and depression in adolescents and young people during and post-pandemic have been widely reported [3,4,5]. Stress, social problems, sleep disturbances, loneliness, and psychotic symptoms also reportedly increased [6,7]. Research has highlighted the detrimental role of home-schooling and social isolation, with technology playing a critical role in daily life [3,8,9,10,11]. However, longitudinal research in adults has not consistently reported negative mental health impacts [12,13]. Further, systematic reviews and mixed methods research have suggested that the effects of the pandemic were not unilaterally negative, highlighting some positives for young people, such as new opportunities for connection, self-care, and reflection [3,5,10]. The range of estimates, differences between populations and subgroups, and mixed experiences in these reports suggest that the psychological impact of the COVID-19 pandemic on young people may be nuanced and complex. Understanding these impacts and their key contributors will enable us to better support those affected moving forward and inform best practices for supporting young people in collective crises (e.g., pandemics, environmental disasters) or under comparable personal circumstances (e.g., incarcerated youth, young people isolated due to illness).

Some research has evaluated young people’s participation in artistic activities as a means of understanding pandemic experiences and/or as an intervention [14,15,16,17,18,19]. In one study, 9–11-year-olds in India created artwork about the pandemic, resulting in cross-artwork themes of positive experiences, negative experiences, unity, safety, hope, uncertainty, gratitude, faith, and future expectations [14]. A study of 21–35-year-olds in Hungary who completed art therapy tasks during the pandemic reported themes of frustration, isolation, loss of control, and support [15]. Taking part was reported to have benefits for expressing and processing experiences [15]. A study in Iraqi Kurdistan asked children aged 6–13 to draw and paint their reflections and responses to the pandemic, highlighting feelings of distress, loneliness, and fear [16]. There is also evidence for the mental health benefits of arts-based therapies for children and adolescents in communities affected by the COVID-19 pandemic and the Liberian Ebola epidemic [17,18,19]. Therefore, research suggests artistic and creative activities have the potential to help young people express and process pandemic-related distress and improve their understanding of it.

However, to our knowledge, research exploring the impact of COVID-19 on mental health through artistic and creative activities has not been conducted with adolescents. This is important because being at this life stage may have presented unique stressors. Further, prior studies of the COVID-19 pandemic have not combined analysis of creative outputs with verbal accounts, each of which offers a different viewpoint, the combination of which may provide access to a wider range of insights. Mixed methods and qualitative studies nested within cohorts also offer much potential for understanding the impacts of the pandemic on different people by allowing researchers to contextualise, reflect, and explore nuances in epidemiological findings. In the London-based longitudinal cohort Study of Adolescents, Cognition and Mobile Phones (SCAMP), participants were found to have increased levels of depression and anxiety compared to pre-pandemic levels [11]. Comparing survey measures taken before the pandemic with measures during the pandemic, new incident depression and anxiety were more common in girls, and new incident depression was more common in participants who had higher phone use and poorer sleep quality at earlier timepoints [11]. Qualitative research in this cohort offers the opportunity to enrich this quantitative research by revealing in greater depth and detail young people’s experiences of the COVID-19 pandemic and lockdowns.

Therefore, this study aimed to uncover a nuanced understanding of SCAMP participants’ experiences of the COVID-19 pandemic using creative activities and interviews, particularly focusing on their mental health and wellbeing. Co-produced research methods and analyses were conducted to ensure the approach and findings were suitable, credible, impactful, insightful, and relevant for young people.

## 2. Materials and Methods

This report has been prepared in accordance with the Consolidated criteria for reporting qualitative studies (COREQ): a 32-item checklist (see Appendix A) and the Guidance for Reporting Involvement of Patients and the Public 2-Short Form (GRIPP2-SF, Appendix A). SCAMP is a London-based adolescent longitudinal cohort study established in 2014 to investigate the relationship between mobile phone use and cognitive development. At study inception, year 7 students (aged 11–12) completed standardised questionnaires and cognitive tasks, with follow-up in years 9–10 (age 13–15) and during college/sixth form (age 16–18). Before the age of 16, informed consent was provided by parents, and after the age of 16, informed consent was provided by participants themselves. To date, SCAMP has recruited ~10,000 adolescents (now 16–23 years old) from 55 schools. It has high socioeconomic and ethnic diversity and represents a wide range of demographics and communities in Greater London and the surrounding areas [20]. With a sub-sample of the SCAMP cohort (see Section 2.2), we conducted a qualitative study to understand their experiences of the COVID-19 pandemic and to contextualise and elucidate prior quantitative findings [11]. This included creative workshops and one-to-one interviews. The North-West Haydock Research Ethics Committee approved the overall SCAMP study and this specific study as an amendment (#14/NW/0347).

### 2.1. Patient and Public Involvement (PPI) and Research Design

SCAMP’s Young People’s advisory Group (YPAG) included (at the time of this specific study) 29 paid members aged 16–20 (72.4% female). This group informs all aspects of the longitudinal study through a variety of regular meetings, workshops, and assignments, which different members sign up/self-select for based on interest and availability. They represent the wider cohort, having all participated in SCAMP at a range of secondary schools in or around Greater London. Ten YPAG members signed up for and contributed to the study design, analysis, or dissemination of this particular study. Firstly, nine YPAG members attended two 2 h online research design workshops in June 2021 with RT and LB to refine the research focus and co-design the methodology. These involved discussing what YPAG members associated with the lockdown, any salient memories, and the key factors impacting their wellbeing and mental health at this time. In the first workshop, the group drafted a topic guide and interview questions to access these key domains. In the second workshop, the group generated ideas for how young people might express themselves creatively, shortlisting some appealing activities for accessing and expressing pandemic experiences and identifying which aspects of the pandemic creative activities should focus on.

Subsequently, we included these workshopped ideas in an online advert calling for creative facilitators to propose a group activity for young people to express and explore their COVID-19 experiences, which received 33 responses. RT and LB first shortlisted these proposals based on facilitator experience with young people, clarity of concept, feasibility, and applicability to research. The 13 shortlisted proposals were then rated by four YPAG members on the following domains: appeal, ability to express themselves, relevance to their experience, and feasibility. The proposal rated most highly by the YPAG was selected, and the workshop protocol was further developed by the facilitator (MW, MPhil Cantab, independent consultant/creative researcher and facilitator, woman) with researcher input (LB and RT). Two YPAG members (AR, RM) also contributed to the qualitative analysis (see Section 2.5 for more detail). AR and RM also contributed to the preparation of this manuscript. All YPAG involvement was supported and compensated according to NIHR guidelines.

In addition, SCAMP’s 2022 summer work experience cohort included 20 work experience students from SCAMP schools (age 16–18) who contributed to qualitative thematic analysis of artworks during their work experience placement at Imperial College London (see Section 2.5 for more detail).

### 2.2. Participants and Recruitment

In 2020–2021, during the COVID-19 pandemic and lockdowns, SCAMP participants were surveyed about their COVID-19 experience, demographics, lifestyle, mobile phone use, cognition, and mental health [11]. This included the generalised anxiety disorder-7 scale (GAD-7) and the patient health questionnaire-9 (PHQ-9), measuring anxiety and depression symptoms, respectively. Of the 1500 that responded to the COVID-19 survey, we aimed for a purposive sample of 15–30 participants for this study, some with moderately severe or severe levels of anxiety and/or depression symptoms (scoring ≥15 in the survey) and some scoring below this level. We aimed for 15–30 participants, in line with prior research showing that data saturation was likely to be reached between 9 and 17 interviews whilst anticipating some attrition between the workshops and interviews [21]. Participants had to be from the SCAMP cohort and have responded to the COVID-19 questionnaire to be included. All COVID-19 survey respondents were sent an email inviting them to sign up for this study in Summer 2021. Using the online platform Qualtrics, participants read an information sheet, signed a consent form, and signed up for a workshop (choosing a date and time). Participants received a £20 Amazon voucher for each data collection session (workshop or interview) they took part in. There was no established relationship between participants and facilitator/interviewers prior to the study commencement, and participants had no prior knowledge of the researcher’s personal goals or reasons for doing the research, though the remit of the study (understanding young people’s experiences of the COVID-19 pandemic) was stipulated in recruitment.

### 2.3. Creative Workshops (August–September 2021)

A series of 1-h applied theatre and participatory arts workshops took place at a University-run community engagement space [22]. These consisted of between 1–6 participants plus 1 facilitator (MW) and 2 researchers (two of RT, PhD, researcher, woman; LB, BSc, researcher, woman; and RBE, BA, engagement and communications, woman). After introductions, participants were asked to physically act out their everyday lockdown activities. Then, they were given prompts for freewriting (a struggle during lockdown, a moment of joy and hopes of coming out of lockdown). Participants then pulled out three key phrases for inspiration in the final section. Finally, participants produced “My lockdown” canvasses using messy play methods, collages, and a collection of arts and crafts materials. These canvasses were taken away as data for thematic analysis and as outputs to be published. These tasks were also intended to help participants access and reflect on their experiences, which they were subsequently asked about in interviews. Following attendance at this workshop, they were invited to sign up for a 1-h interview slot using the online platform Doodle.

### 2.4. Interviews (September–November 2021)

Semi-structured interviews were conducted via videocall (Microsoft Teams, WhatsApp, or Facetime) or phone call (depending on participant preference) by RT, LB or RBE. At the start of the interview, participants were welcomed, reminded that they could stop at any time and skip any questions, given a summary of the contents and purpose of the interview, and given the opportunity to ask any questions before the recording began. The topic guide covered the following in relation to the pandemic and lockdowns: initial reactions, positive experiences, education, hobbies, relationships, health, loss, differences between first and second lockdowns, coming out of lockdown, and overall reflections. Each interview was completed with a debrief, time for questions, and a handout of sources for support. The full interview topic guide can be found in Appendix A. Interviews were audio recorded and transcribed verbatim by the TakeNote transcription service.

### 2.5. Analysis

The analysis was grounded in interpretivist epistemology, where people’s accounts and researcher interpretations were understood to be subjective and context-dependent whilst capturing experience in a valid and meaningful way [23]. Interviews and artworks were analysed using Qualitative Thematic Analysis, following Braun and Clarke’s 6 phases of analysis [24] and Dewa et al.’s co-production methodology for thematic analysis [25]. To attenuate any biases introduced by analysis of different mediums (verbal accounts and artwork), analyses were conducted separately, bringing results together at the interpretation stage. Code frequency analysis was also used to compare how common different themes were in the interview data of different groups (by sex, school type, and mental health).

#### 2.5.1. Qualitative Thematic Analysis of Interviews (Autumn 2022)

Two members of the SCAMP YPAG (AR, RM) were trained in qualitative thematic analysis by RT. The transcripts were analysed by RT, AR and RM (RT all transcripts, AR 9 transcripts, RM 8 transcripts) using Microsoft Word (highlight and comments functions,) and Excel (collating the coded data according to codes) (both Microsoft 365 MSO, Version 2403, Redmond, WA, USA). Semantic and latent codes and themes were identified to capture young people’s experiences of the pandemic, with a particular focus on their mental health and wellbeing. Validity was ensured by duplicate coding (triangulation), with each interview transcript coded by both RT and either AR or RM. A draft structure of themes was produced by RT, AR and RM, and the complete analysis (data presented within codes within themes) was reviewed by LB and RBE. The thematic structure was then further refined and finalised based on this feedback.

#### 2.5.2. Qualitative Thematic Analysis of Artworks (July 2022)

Artwork was analysed in collaboration with SCAMP’s 2022 work experience cohort through interactive seminars. RT provided workshop-based training on qualitative thematic analysis, adapted from Dewa et al.’s methodology [25]. Each young person contributed up to three codes that represented each artwork using Mentimeter. Codes could designate literal content or underlying meaning. The young people then collectively drafted a thematic structure using Google Jamboard (browser-based digital interactive whiteboard application, Google Workspace, Mountain View, California, United States). RT and two of the students reviewed these codes and themes for referential adequacy, providing definitions and some renaming. The thematic structure (artworks-codes-themes) was then reviewed by AR, RM, LB and RBE before finalisation by RT. These themes were validated for consistency by comparing them with interview transcripts, where participants were asked about the meaning behind their artworks.

#### 2.5.3. Code Frequency Analysis (January 2023)

To explore group-level variation in experiences, the distribution of interview data falling across the different themes was compared. Firstly, the proportion of coded data falling under each subtheme or overarching theme was calculated for every participant (the number of coded segments of data per subtheme was divided by the total number of segments coded). This gave an indication of how much of each interview’s subject matter was applicable to each subtheme. To evaluate whether certain subthemes were touched on more or less by certain groups, the average (mean) proportion of interview data coded to each subtheme was calculated for the following participant groups: boys compared to girls; participants from state compared to independent schools; participants scoring at a moderately severe or severe level of anxiety and/or depression symptoms compared to participants scoring below (according to the PHQ-9 or GAD-7 surveyed during the lockdown period).

## 3. Results

Twenty-one participants took part in the study overall. Four took part in workshops only and were not interviewed, 5 took part in interviews only and did not attend a workshop, and 12 took part in a workshop followed by an interview. In total, 16 attended the creative workshops, and 17 were interviewed. One person signed up but did not show up to either session, for which a reason was not given. No Adverse events occurred during the study. Interviews were between 11 and 56 min in length (mean 33). The sample was aged 16–18 (mean age 17.73), 62% female, 62% state school attendees, and ethnically diverse (Table 1). Thematic analysis of interviews gave rise to two overarching themes, four themes, and 14 sub-themes (Figure 1). Data saturation was reached at interview transcript 12. Thematic analysis of artworks gave rise to four themes and seven sub-themes (Figure 2), and data saturation was not reached. The following sections provide results from the interview analysis, followed by the artwork analysis, by presenting the thematic structure, followed by an explanation and example quotes/artwork for each overarching theme, theme, and subtheme. The italicised text provides direct quotations, with an ellipsis within sections of text indicating that interim text has been removed for brevity.

### 3.1. Qualitative Thematic Analysis of Interviews Results

#### 3.1.1. Overarching Themes

Two themes (restriction and adaptation) were identified as overarching themes because they were especially saliant, present in all interviews, and cut across the remaining themes. Firstly, all aspects of life were restricted in some way. The actions people could take, resources they could access, and the ways in which they could do things were practically limited, which also led to a sense of psychological restriction. This was linked to feelings of isolation, uncertainty, powerlessness, and hopelessness. However, these changes and losses were also discussed in relation to adaptation; people found new and innovative ways of approaching school and work, social connections, hobbies, themselves, and their day-to-day lives. As such, many described finding the second lockdown easier because they had experienced it before, adapted to it, and knew what to expect. Many also drew a new sense of perspective or resilience from these experiences.

“*The biggest change was, yes, restriction. I felt like there were shackles on my feet. I couldn’t do what I wanted to do.*”

“*Everyone had the same thing, where they just, like, stuck in their own little box.”*Participant 15 (P15), male, 16 years

“*I think it’s, kind of, shown how people can adapt to things… that actually, it’s not the end of the world when they change.*”P6, female, 18 years

“*I just thought to myself, like, I‘ve done this before and I‘ve got through it and I can do it again… the end result, I would say my mindset is a lot more positive. In the first lockdown it was very negative, but I turned it around as best that I could.*”P8, female, 17 years

#### 3.1.2. Themes

##### Theme 1: Change

Societal and Social Change

This subtheme encompassed societal and social changes. As well as changes during lockdowns, such as to how people could interact and behave, this included changes that occurred following the lifting of restrictions. For example, loss of social skills, increased social awkwardness, increased attention to personal hygiene and potential transmission of disease, increased home and flexible working, and reduced physical contact between people.

“*It was out of nowhere as well, it was just like on a Friday, you’re not going to go into school anymore, everything is going to be different.*”P2, male, 18 years

“*I also got this thing, vibe from other people that they don’t know how to talk to me. So it, kind of, felt like we were all in the same situation of not knowing how to talk to people.*”P8, female, 17 years

“*You’re used to eating without sanitising, sometimes you’d just eat, just outside, wherever. Now you’re like, How did I do that in the past?’…I’m disinfecting all the tables, I‘m disinfecting my hands. It’s things that I used to never do before that now I do. I‘m like, how did I not do it in the past*?”P9, female, 16 years

Personal Journey and Development

This subtheme is about personal change from the pandemic. Some participants talked about being less close with friends, less able to connect with others, or changing their future expectations following disruptions to education and other opportunities. Yet, many also talked about a renewed appreciation of their loved ones, freedom, health, and themselves. Many participants found independence and self-reliance and learnt they could emerge strongly from difficult experiences.

“*I would have to make friends all over again with people that I used to be friends with, because I stopped talking to them. It was like we were all completely different people now, because we experienced something really influential on our lives, so we all either took it very negatively or very positively and that, kind of, made us the way we are today post-lockdown.*”P8, female, 17 years

“*Anything I‘ve gained, yes, quite a lot of lessons. I‘ve seen a side of me that I‘ve never seen before or experienced. I‘ve gained a lot of courage and I would say humanity to be honest. That’s the main thing.*”P15, male, 16 years

Lifting of Lockdown

The lifting of lockdown measures had a variable impact: some participants found it easy to adjust back and adopt previous routines, finding it liberating, exciting, or hopeful, whilst others found it challenging to socialise and reintegrate into society, finding crowds and in-person meetings overwhelming.

“*It was the biggest relief, to be honest. It felt a lot more like suddenly there is an end in sight, when you’re giving a specific date of, like, okay, so now this will happen,’ it just suddenly feels like, okay, the world’s not ending.*”P4, female, 16 years

“*I think I was more scared about coming out of my shell, because I think, you know, the way I‘d isolated myself, it was scary going back and talking to those people. Explaining why I wasn’t online. I think that was scary… I think I was so comfortable to this type of lifestyle, that going out was just tiring for me.*”P17, female, 17 years

“*It sounds weird, but I was, like, really scared of going into the supermarket after the pandemic. It’s just, like, seeing that many people in a small space.*”P13, female, 17 years

Uncertainty (shared with challenges, below)

A subtheme shared by the themes of “change” and “challenges” was “uncertainty”. The level of uncertainty about what would happen, particularly around education, exams, and grades, left many young people feeling unsupported and anxious.

“*When lockdown first happened it was a bit of a shock to me because I was, like, wow, what happens next? I mean, what is going on?*”P15, male, 16 years

“*Over a period I would say your brain had started questioning, like. It was very strange, that questioning your brain was doing. It was, like, oh, will this violence ever end?’ Or, what if lockdown lasts for years on top of years? What’s going to happen to my education?*”P14, male, 18 years

“*It was just a really big grey area, so there was no certainty. And there was just constant stress.*”P8, female, 17 years

Anxiety (shared with challenges, below)

Anxiety was another subtheme shared by the themes “change” and “challenges”. Predominant sources of anxiety were change, uncertainty, fear of infection, and fear of infecting other people. However, the largest source of anxiety was school and education-related anxiety: fear of what would happen with exams and grades, falling behind, or not doing as well due to remote education. Most of the young people found the second lockdown easier than the first, which was attributed to the fact that schools were more prepared, so they felt more supported and had less educational stress. In the second lockdown, there was also a sense of “having done this before,” which reduced uncertainty and anxiety.

“*Especially the first lockdown. I was, like, what is going to happen next?’ I mean, cases were rising, there was panic everywhere. I wasn’t, you know, in a good place at the time and I was just confused about the future and for me especially there was just a big question mark on the future. I mean, what is going to happen next? What is this?*”P15, male, 16 years

“*The problem was we were all constantly worrying about our future, especially with loads of people’s work experience got cancelled. But when lockdown was actually announced, I think, yes, there was no support from the school as well, which is terrible. I mean, we did have exams. We knew we were going to have exams by the end of it, but there was no support from school. So all of that pressure really just got to us.*”P17, female, 17 years

“*I guess my biggest concern was just if it was ever going to end, you know? Before we got any dates, before we got any roadmaps, any tiers, any of that, it just felt like never ending and you’d open the news, and you look at the news, you look at the statistics and every day it just seemed more and more negative and depressing. And my biggest concern was just that, that it was never-ending and that I was going to spend my whole life in lockdown.*”P4, female, 16 years

##### Theme 2: Challenges

Uncertainty (shared with change, as above)Anxiety (shared with change, as above)Loss

As a subtheme, loss included bereavement but also loss of freedom, opportunities, and connections. Even for those without a direct bereavement, having friends who had been bereaved and the potential of bereavement at any time was a source of distress.

“*He was just there and then he wasn’t the next moment. So, it just proved to me that things do change overnight sometimes.*”P5, female, 18 years

“*I was just thinking ‘Now, please, I just don’t want them to get COVID.’ Because if anything happens, I can’t really bear another loss.*”P17, female, 17 years

“*Even though it wasn’t me directly, just knowing that people actually are losing people they love due to the pandemic, it makes me feel like I‘ve lost something as well, even though it wasn’t me, you know.*”P8, female, 17 years

Hopelessness and meaninglessness

Although many young people felt initially excited about lockdown due to the novelty of getting to stay at home and have a break, this was not maintained. For many, the monotony, lack of day-to-day routine and purpose, without a clear end point in sight, led to feelings of hopelessness and meaninglessness. Those who found the second lockdown more difficult than the first attributed this to lost novelty and feelings like frustration and boredom.

“*I wasn’t properly getting up for school. I wasn’t really getting dressed, it just wasn’t good because I was so, like, stuck in the fact that none of it really made sense, I didn’t really have a reason to.*”P4, female, 16 years

“*During the day I was just bored, and the boredom that I felt, not every day but almost every day, it just got repetitive and I was just bored of being bored, that it almost got me annoyed that I was bored, I just didn’t know what to do.*”P8, female, 17 years

“*Is it really worth it? Is it really worth me doing all this work if no one’s really checking it and no one’s really doing anything?’ I would say it was pretty bad, if that makes sense. I wasn’t really stable in the first couple of weeks. Yes, it was just being indoors…You want to do something else. It affects you mentally.*”P14, male, 18 years

“*I thought, yes, this is it, this is how the world ends.’ I was, like, there’s no way out’.*”P15, male, 16 years

Loneliness and disconnection

Relationships with family and friends and levels of emotional support and connection seemed to be key factors in determining whether people’s experiences were positive or negative. For those who drifted apart from their friends, clashed with their families, or did not feel particularly connected to others, the lockdown period was extremely difficult.

“*It just started to really hit me hard, like, how lonely, obviously lockdown can be… I think it definitely worsened myself and a lot of other people’s mental health because you’re isolated, when you’re alone, you tend to get into your own head and you don’t have, you know, space to get that out*”P4, female, 16 years

“*I was affected in terms of not being able to see so many people, so I wasn’t probably able to say everything I wanted, or maybe just to reflect myself…You weren’t able to express yourself fully.*”P9, female, 16 years

“*Just didn’t want to talk to anybody and, even at home, we’d just have constant fights, you know. Yes, we’d just have constant fights… There’s never been a moment that all of us have spent so much time together and, yes, it was just everyday there was something to clash about.*”P17, female, 17 years

Online education (shared with lockdown life, below)

The most prominent source of stress and anxiety was online education. Most participants felt that lessons, resources, and support from schools were inadequate at first and that they were missing out on their education. Many discussed a contrast between the first and second lockdowns. Uncertainty around exams, grades, cancellations and changes to policy was very unsettling and stressful, with participants not knowing what would happen, how to prepare, or if their performance or prospects might be negatively affected by the changes set out. Several found it hard to work at home, particularly if they were sharing a smaller space with many of family members and siblings. By the second lockdown, some schools and teachers were more prepared and offered comprehensive lessons and a structured timetable, promoting a positive sense of routine and opportunity to progress.

“*I feel like my mental health was affected the most around the time of my exams. Because I felt quite overwhelmed, especially not having support and not having teachers right there to help you.”*P2, male, 18 years

“*My school was really terrible at doing that; they didn’t really care. And I didn’t care either, to be honest. They didn’t send anything. I didn’t care.*”P17, female, 17 years

“*I have other siblings, they just come in and out my room. My room upstairs is, like, the living room upstairs because it has a TV and stuff, so people just come in, they chill, and it’s like I‘m never alone to focus. There’s always something happening.*”P12, male, 17 years

“*I feel like the second and third lockdown, the school had, kind of, figured it out by then and realised how the system works, and all the teachers were more into it. In the first lockdown, no one knew how anything worked, like, the teachers didn’t know, we didn’t know*”P13, female, 17 years

##### Theme 3: Lockdown Life

Online education (shared with challenges, as above)Hobbies and technology

Technology featured heavily in people’s accounts of the pandemic. Most people consumed many media during lockdown, such as streaming TV shows and films and playing videogames. Participants sometimes described the impact of technology negatively in terms of aimless scrolling, excessive screentime, and making unfavourable comparisons between their lives in lockdown and others based on social media. However, technology also created opportunities to connect with loved ones, continue their education, and engage with new hobbies or return to old hobbies (such as baking, sewing, writing, and exercise). Some participants described friendships being enhanced by the ways they could connect using digital technology.

“*When I wasn’t studying or revising, I would, like I said, pamper myself, watch stuff on Netflix, but also just FaceTiming a lot of people. I feel like I ended up talking a lot more with my friends because we made that effort to, like, you know, do house parties and Zoom meetings and things like that.*”P11, female, 18 years

“*Spent time on my phone on social media most of the time. I didn’t really do anything productive either. Literally, every day, I think I would sleep for most of the day, and then wake up really late, stay on my phone, and then sleep, and that was my day.*”P17, female, 17 years

“*I didn’t find it too bad because most of it was, kind of, screen time for a purpose, and what I find, what I enjoy least and find affects my mood most in terms of screen time, is idly scrolling through things or procrastinating stuff.*”P6, female, 18 years

Health and lifestyle

Most people felt their health-related behaviours changed. Most reported a negative impact on their sleep routines, typically going to bed late and sleeping in later. Many also exercised less, ate more unhealthily, and did not leave the house often. However, some people started exercising more, eating more healthily, and spending more time in nature. Having a positive routine that felt productive and meaningful was conducive to feeling better mentally and physically.

“*I think my health, probably was not the best in lockdown, both mental and physical. I just wasn’t really exercising very much, I wasn’t really doing much. I was stuck in bed a lot, I couldn’t really find the energy to get out of bed, despite not using much energy.*”P4, female, 16 years

“*Every day I was really just sitting inside sleeping. Not eating enough. Not staying healthy… loads of people said they were going to learn a new skill or, you know, exercise, or just change themselves, and there’s pressure to do that as well, because I just sat down every day on my phone. I didn’t learn a new skill and actually became more unhealthy.*”P17, female, 17 years

“*I got into some healthier habits because I, like, I guess being at home around someone all the time, obviously parents, they’re just there, I guess they make sure you’ve got healthy habits. So, like, I had good meals for lunch and dinner and things, as opposed to just snacking between and stuff like that, and I slept pretty good. Yes, I feel like I was pretty healthy during lockdown, especially at the start when it felt like a lot of time we made effort to go out and get some fresh air*”P6, female, 18 years

Connection and support (shared with overcoming adversity, below)

The availability of connection and support from family, friends, schoolteachers, and mental health professionals made a big difference to people’s experiences and mental health during lockdowns. Relationships within cohabiting families seemed especially important; those who got on well often enjoyed increased quality time, building closeness, and sharing routines, whereas those who clashed with family had a much more stressful and isolating experience.

“*You know how on social media where you can just interact with, like, other people and just speak to random people about stuff? I think that was pretty good… I went on, like, TikTok and just spoke to other people, you know? So, that was, kind of, it helped stress relief.*”P10, male, 17 years

“*I do think a lot of the family relationships felt closer, which is nice. I think, with my mum, it was kind of, a natural product of just being together a lot of the time, getting on well. With my dad, it felt I had to put in more effort to make that happen, but I think we did, and so that worked really well.*”P6, female, 18 years

##### Theme 4: Overcoming Adversity

Connection and support (shared with lockdown life; see above)Resilience

Many young people developed a new sense of resilience during this experience. Having to undergo major changes in such challenging circumstances gave them confidence in their ability to cope with future difficulties.

“*With the first lockdown you probably thought it was forever, because it has never happened before. But with the second lockdown I knew it was going to end… You do get that feeling at the beginning, that, oh no, it’s happened again.’ But then you just carry on and then at the end of the day there is light.*”P9, female, 16 years

“*I feel I have built up my resilience. All those things that just suddenly happened.*”P7, female, 18 years

“*On a personal level, I feel like I‘ve learned to do things on my own, rather than sit around waiting for someone else to do it for me.*”P2, male, 18 years

Self-care and fulfilment

Having more time for self-care and seeking out sources of everyday fulfilment benefited some participants’ wellbeing. This included taking walks, spending time in nature, relaxing and enjoying a slower pace of life. Some young people even found staying at home less stressful than their day-to-day lives at school.

“*I think, yes, definitely going out, you know, to the park, and then having time for myself, on my own, if that makes sense. Just alone time, because I think everything was moving too fast before. You know, but in that time I got to spend time with just myself, alone, without friends. Without anybody. Just sitting there alone, which I think I, kind of, yes, I think that was needed.*”P17, female, 17 years

“*I think everyone went though, like, a baking phase and I was one of them. I spent a lot more time in the garden, like, I think I appreciated my garden more. I watched a lot more things on Netflix, which was nice. Dyed my hair. So, yes, it was really nice because I feel like I got to, like, pamper myself a lot more too.*”P11, female, 18 years

### 3.2. Qualitative Thematic Analysis of Artwork Results

Although no exact themes were shared by both thematic structures, the themes “trapped”, “technology”, “loneliness”, and “overwhelming” show similarities to the themes “restriction”, “hobbies and technology”, “loneliness and disconnection”, and “anxiety” from interviews. This suggests these concepts were at the forefront of participants’ minds when they reflected on the pandemic in both modalities. Although the interviews were able to draw out many positives through detailed and extensive accounts, the artworks captured overall experiences “in a snapshot” and appeared to convey more challenging than positive feelings. This suggests that the experience was predominantly difficult for most participants.
“*I, kind of, did portray it as very negative, I realised that after I‘d finished it. It wasn’t all negative, but when I think of lockdown my first thoughts do go to the more bad side rather than the good side, because the negatives did outweigh the positives.*”P8, female, 17 years

#### 3.2.1. Themes

##### Theme 1. Trapped



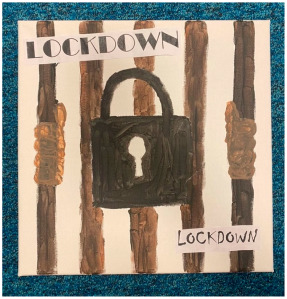



Many artworks depicted being caged or locked in. This might symbolise both the physical restrictions in place and feeling psychologically trapped and contained, which was echoed in interviews.
“*I painted a padlock, representing lockdown, feeling trapped; you’re staying in one place, we’re all, kind of, trapped, we’re all essentially locked. So, I literally just expressed that, just did that.*”P12, male, 17 years
Loneliness (shared with negative mental wellbeing, below)

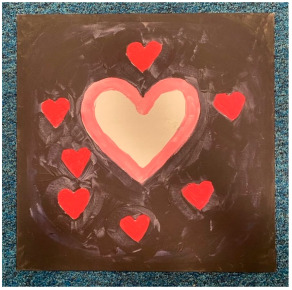



Shared by the themes “trapped” and “negative mental wellbeing”, loneliness and isolation were subthemes found in many artworks. Loneliness was also reported extensively in interviews.
“*In my painting I drew a heart on the left side as it was really close to my heart, but there was just this tiny little gap in the middle, and that one heart, kind of, represented my sister, who is really close to me, literally a blood tie, but I couldn’t see her, and I couldn’t talk to her, I couldn’t really hug her, she was so close and yet so far. That’s what I felt in lockdown.*”P8, female, 17 years
Overwhelming (shared with negative mental wellbeing, below)

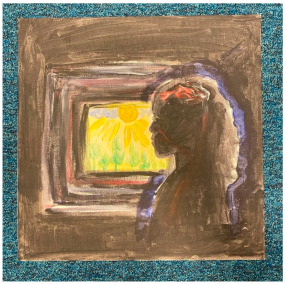



Shared by the “trapped” and “negative mental wellbeing” themes, much of the artwork conveyed a sense of overwhelm. This subtheme captured intense emotions and a sense of engulfment, which also came across in interviews. The lack of support described by some participants, such as not being able to talk about their feelings whilst stuck at home, is likely to have contributed to this sense of overwhelm.
Catastrophic (shared with negative mental wellbeing, below)

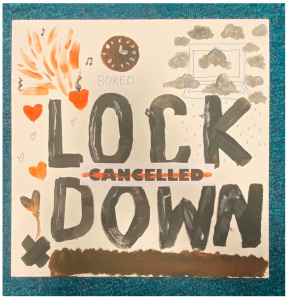



Many artworks conveyed a sense of catastrophe, of momentous, dangerous, and tragic events. Although day-to-day living was described as mundane and uneventful in interviews, the wider context meant many felt like they were in the middle of something historic and apocalyptic at the same time.

##### Theme 2. Negative Mental Wellbeing

The negative mental wellbeing theme shared the subthemes of “loneliness”, “overwhelm”, and “catastrophic” with the theme “trapped” (as above), as well as including the subthemes “disorientation” and “contrast”.
Disorientation

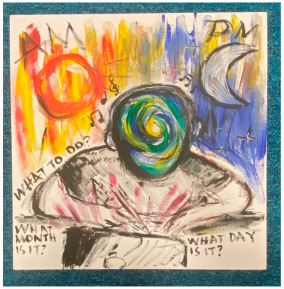



Many artworks conveyed a sense of disorientation. The high levels of change, uncertainty, stress, lack of routine, repetitive days, isolation, and time to ruminate expressed in interviews are all plausible contributors to a sense of disorientation.
“*I had the sun on one side and the moon on the other side, showing that the days pass and you just sit there indoors, and you don’t really realise what’s happening in the outside world because you’re just stuck at home.*”P13, female, 17 years
Contrast (shared with positive emotions, below)

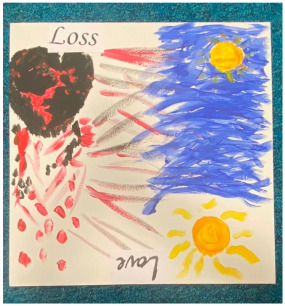



Many artworks showed “good” and “bad” together, with some deliberately splitting the artwork in two to show a direct contrast. Contrast was not in the thematic structure coming out of interviews, but the interviews frequently contrasted positive with negative experiences. When reflecting on their experience, many participants talked about how there were good and bad elements to it or how they took learnings and growth from negative experiences.
“*What I created was a painting where it’s, kind of, split in the middle diagonally, and one side, so, the left side, the bottom side-, no, I mean, the right side, the bottom side, is, basically, the bad things that came from COVID-19 and stuff like that. And the up side was the good side.*”P5, female, 18 years

##### Theme 3. Positive Emotions

The theme of positive emotions included the subtheme of contrast (shared with negative mental wellbeing, see above), as well as “serenity” and “hope”
Serenity

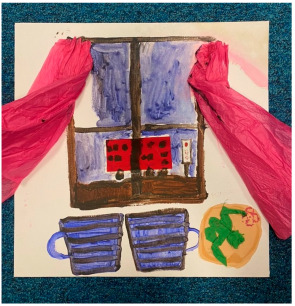



Lockdown gave many people time to reflect, enjoy companionship, experience gratitude, and enjoy a peaceful, relaxing, and comfortable day-to-day life. The artworks show activities such as drinking tea and spending time with nature, representative of the slower pace of life and calm which some of the young people discussed in interviews.
“*I think that’s what I remember from lockdown especially. Going out, you know, in the park having a walk. So I created that… For me, it was I think one of the best moments of lockdown, getting to go out, you know. Going to that park. Fresh air. I think that was the best part.*”P17, female, 17 years
Hope

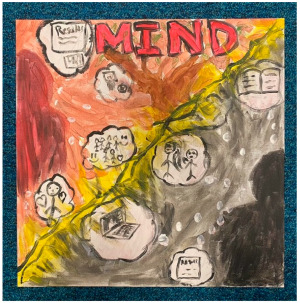



Amongst the challenges, hope also came across thematically in the artworks. In interviews, hope was often mentioned as a motivator to keep going, although hopelessness also came up frequently.
“*I coloured it yellow because I wanted to indicate even though you’re experiencing bad stuff and good stuff there’s still light in you that you can change, so, that’s why.*”P5, female, 18 years
“*So, one half was really dark and I used really dark images and colours and it was just grief and loss, but the other half was a bit of hope because when COVID-, well, lockdown ended, I could see my friends and I thought, like, there’s always light at the end of a tunnel I thought I‘d just get two sides, two ends of the spectrum onto one bit.*”P15, male, 16 years

##### Theme 4. Technology



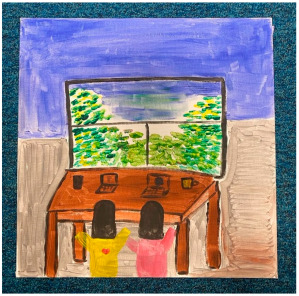



This theme encapsulates the technology depicted in many artworks- specifically computers and gaming consoles. This was not always depicted negatively, but time spent on screens was clearly salient when people reflected on their lockdowns. Some of the artwork implied that people felt like they were living in a virtual reality and detached from the real physical world.
“*I drew a picture, it’s all, like, a virtual world, because you couldn’t really experience the real world anymore, because you’re pretty much stuck. Being indoors was, you know, mentally difficult. It was mentally sort of like helps the illusion that I‘m actually outside, so I use the headset, you know, play a couple of games, which gives you the illusion that you’re actually in the outside world. So, it made me feel stable. So, on my canvas I drew a picture of a virtual world with my headset*”P14, male, 18 years

### 3.3. Code Frequency Analysis Results

The prevalence of each subtheme and overarching theme within the interview transcripts was averaged at the group level by gender, mental health, and school type, as seen in Figure 3, Figure 4 and Figure 5 (as data could be coded under more than one subtheme and/or overarching theme, percentages do not sum to 100). The themes more predominant in those with lower anxiety and/or depression symptoms (Figure 3) seem to reflect the more positive, reflective, and adaptive aspects of lockdown, whereas the more limiting aspects of lockdown are represented more for those with high symptoms of depression and/or anxiety (as well as, unsurprisingly, themes reflecting more negative emotions). Yet, whether these factors lead to better mental health, arose from better mental health, were remembered more positively by people with better mental health, or some combination cannot be ascertained from these data. One gender did not appear to discuss more challenges or “negative” themes than the other (Figure 4), but different challenges seemed to be experienced by boys and girls. The themes predominant in the experiences of independent school students were much more positive (Figure 5). It is plausible that, as an indicator of socioeconomic status, independent school students would have had more physical space, higher quality neighbourhoods to explore, and less familial stress than those at state schools. For example, independent school students talked more about self-care and fulfilment, which is enabled by higher resources and better-quality environments. Also, crucially, higher-resource schools are likely to be able to provide a better online education, which was the most widely discussed stressor.

## 4. Discussion

This study invited 16–18-year-olds from the London-based SCAMP cohort to share their experiences of the COVID-19 pandemic through interviews and artwork. Through the themes of adaptation, restriction, change, challenges, negative mental wellbeing, overcoming adversity, lockdown life, trapped, negative mental wellbeing, positive emotions, and technology, this research provided rich insights into what life was like for young people during this time, how they felt, and what factors impacted their mental health.

Qualitative insights from the present study can help elucidate the partial picture generated by prior quantitative research in the cohort [11]. Statistical analysis of survey data identified high and increased rates of anxiety and depression during the pandemic, especially for girls, people with prior sleep difficulties, and people with high prior technology use [11]. In the present study, girls discussed more loneliness, isolation, hopelessness, and meaninglessness than boys, which may have contributed to their mental health. However, girls also talked more about self-care, fulfilment and personal development. Whilst boys discussed anxiety more, they also discussed connection and support, hobbies and technology, and adaptation more. Potentially, how boys socialised, passed the time, and found support was more accessible through online and remote mediums, meaning girls’ mental health was more badly affected. However, in the quantitative research, girls scored more highly and showed a bigger increase in anxiety symptoms during COVID [11], whereas boys’ qualitative data were more often coded under the “anxiety” subtheme. This may highlight a discrepancy between psychiatric anxiety symptoms (measured quantitatively) and anxiety, as discussed in conversation, which can refer more broadly to a range of clinical and non-clinical experiences.

In quantitative research with this cohort, moderate social network site users had slightly lower odds of new incident depression during the pandemic [11]. This was surprising because pre-COVID total mobile phone use increased risk, and earlier research in this cohort has found detrimental associations between social media use and mental health [26]. However, in the qualitative study, those with lower anxiety and depression talked more about hobbies, technology, and social connections. Further, many young people shared cherished memories of virtual activities shared with friends during lockdowns. This all suggests that social media use during this time could have enabled connection and mitigated loneliness for some people. The mixed role of technology, as both extremely useful and potentially harmful, echoes prior research [27,28]. Passive technology use without a clear purpose (such as aimless scrolling, social comparison, and streaming TV shows all day) tended to be described as detrimental, whereas technology used intentionally, such as for studying, socialising and accessing hobbies, was not presented as problematic. Therefore, complexity should be acknowledged when discussing digital technology use and mental health, as opposed to representing it as uniformly harmful. Sleep disruption and unhealthy sleep behaviours were also widely discussed (and have been reported in other research [4,29]), which is consistent with the quantitative finding that prior sleep problems were associated with worse mental health in COVID; prior sleep issues could have been exacerbated by the pandemic (e.g., by changes to routine), thus negatively impacting mental health.

Analysis of interviews and artworks showed that there was a wide range of experiences, some predominantly negative and some more positive. A study of adults with mental health problems in the UK also reported a wide range of experiences, with some people finding the pandemic extremely challenging and some enjoying the respite [30]. Echoing our findings, this study also identified uncertainty and isolation as detrimental to mental health and hobbies, social support, and connection as beneficial [30]. It is a strength of a qualitative approach that it can cover a range of emotions and experiences at once. A single numerical value indexing mental health or wellbeing at one point in time will not capture all the nuances we can observe qualitatively. A range of positive and negative experiences, between or within individuals, could effectively “cancel each other out” on average. For example, a systematic review and meta-analysis of mental health symptoms before and after COVID-19 in all populations found no significant change in general mental health or anxiety and a very small increase in depression symptoms [12]. However, some of our participants reported less anxiety because external stressors were taken away, and some reported increased anxiety due to factors like online education and uncertainty. If some people coped well or even benefitted, this could obscure those who had difficulties, but it does not diminish the significance of these difficulties. Additionally, many individuals reported a mix of challenges and positive experiences, and many artworks depicted “good” and “bad” together, which one quantitative measurement of their mental health would not capture. Stratified analyses [11,12] and measurement of multiple domains of mental health [31] can enable quantitative research to better unpack the complex patterns occurring within large-scale events over time.

### 4.1. Strengths and Limitations

This study has many strengths. Firstly, being nested within a larger quantitative cohort study provided deeper interpretation and insight, better contextualisation, and a more holistic picture than studies that have utilised purely quantitative or qualitative approaches. The combination of interviews with creative practice uncovered more themes than either method alone, suggesting a wider range of lived experiences and impacts were made accessible by combining methods. Potentially, this gave students whose communication styles are less suited to a formal interview a better opportunity to express themselves [32]. Working with young people to design the approach, methods, and protocols may also have helped create an effective and comfortable space for young people to express themselves and reflect on their experiences. Many students reported that the workshops, interviews, or both were beneficial because they had not yet had the opportunity to process their experiences. Many also reported that the workshop was fun and enjoyable. Future research employing this approach could evaluate it as an intervention in addition to a research methodology. Further, the involvement of young people in analysis drew out different codes and themes than those identified by RT alone. Their shared experiences and expertise in how young people express themselves may have improved our understanding of what was being conveyed. The co-researchers involved in the analysis also reported personal benefits such as gaining research experience, research skills, confidence, communication skills, organisational skills, and widened perspectives. The sample, like the cohort and city it represents, was ethnically and socioeconomically diverse.

In terms of limitations, this sample was diverse but not representative of the cohort or young people in London in every way; for example, there were no black participants. Although quantitative differences in mental health were not observed by ethnicity during the pandemic [11], we have demonstrated that qualitative research can unveil new findings (e.g., differences between independent and non-independent school students not observed quantitatively [11]). During the pandemic, black ethnicity was associated with a higher risk of COVID-19 infection and mortality, so the impact on young black people and families is important to understand in future research [33,34]. Additionally, whilst a strength in terms of representation, high ethnic diversity could have impacted the cohesion of results, given the potential impact of cultural factors on people’s experiences [35,36]. Further, combining results from fundamentally different mediums (artwork and verbal accounts) could introduce bias due to incomparability. However, conducting analyses separately should have prevented the conflation of findings derived from different processes. Further, having results from both mediums gave access to a wider range of themes, potentially attenuating some of the biases present in each (e.g., a broad interview could potentially dilute or distract from the most important messages, and artworks could fail to capture the whole picture). Although we combined analysis of images and interview transcripts, direct analysis of recordings of the workshops and/or interviews could have provided even more information or aided interpretation, e.g., picking up on body language or verbal cues [37,38]. Finally, it is possible that the personal expectations and biases of those designing the methodology or those analysing data could have influenced the themes that arose. However, the involvement of multiple coders and the inclusion of young people should have reduced the degree of bias and increased credibility. Additionally, if the themes were impacted by the co-researchers’ own experiences and expectations, these also reflect the remit of the research, i.e., what was young people’s experience of the pandemic, and how was their mental health affected? To the extent that these accounts are seen as co-constructed by researchers and participants, leaning the research process heavily upon co-researcher involvement bases findings on the target populations’ experiences from both directions.

### 4.2. Implications

In terms of implications for policy and practice, this study has shown the importance of wider determinants of mental health; when addressing young people’s mental health, it is crucial to look at their life context, such as their environment and day-to-day activities. In terms of good mental health, this study has highlighted the importance of young people feeling a sense of control over their life and their future, which can be promoted through educational, social, and economic support and opportunities and through therapeutic techniques to enhance self-efficacy, resilience, and adaptability. It is important that young people’s agency is also enhanced by including them in the decision-making that influences them. These interviews strongly highlighted the issue of educational injustice, with poor quality remote provisions and uncertainties in education creating a huge amount of stress and anxiety. This issue has been identified in prior research in the UK context [39], and the disruptions experienced during lockdown may have exacerbated existing trends of increasing school pressure [40]. Additionally, the quality of educational support students experienced varied, with independent school students discussing challenges with online education less. Students would benefit from teacher training that raises awareness about how to avoid, identify, and address educational stress in the classroom. The Royal Society of Medicine recently called for a greater amount of UK teacher training on mental health; despite being the source of contact most called upon for child and adolescent mental health support, just 40% of teachers feel equipped to teach students with mental health needs [41]. Although funding has increased for school-based mental health support in England (funding is now available for one staff member per school to be trained as a senior mental health lead, and in-school NHS mental health support covers 3000 schools [42,43]), it is still not sufficient to meet the mental health needs encountered in every classroom [41]. Participants in this study also reported uncertainty around changing public health policies and the information provided about the pandemic, highlighting the importance of clear communication and the establishment of trust between young people and decision-makers. Relationships at home, the physical quality of the home environment, the importance of social connection, and access to nature, hobbies, and self-care were also important. Having routine and structure, sometimes enabled by support from school and/or family members, was also beneficial. All these factors should be considered when supporting young people in general, particularly in response to future pandemics or similar limitations on a smaller scale (e.g., incarcerated youth or young people with health conditions that limit their independence or mobility). This necessitates that psychiatric professionals, educational professionals, social workers, and local authorities are given the infrastructure and resources to work together rather than in silos such that they can holistically consider and address the way that factors on different levels collectively determine young people’s mental health.

## 5. Conclusions

This qualitative study has uncovered unique insights into the impact of the COVID-19 pandemic beyond what has been observed quantitatively in this and other cohorts, shedding light on reported changes in mental health and differences between groups. This work suggests that young people had a wide range of experiences during the COVID-19 pandemic, with many facing challenges with online education, loneliness, uncertainty, and restriction. It has highlighted the benefits of good connections, emotional support, good opportunities, a safe and secure home environment, healthy habits, meaningful and enjoyable hobbies, and hope, gratitude, and self-belief for fostering adaptability and resilience. These findings support a systems-level public health approach to young people’s mental health and have important implications for how we approach contexts of change, current events, and isolation as they impact young people on different scales. This methodology was extremely effective at creating a beneficial research process for participants and generating a holistic picture of their experiences, highlighting the importance of creative approaches, involving young people, and mixed methods in public health research. Therefore, these methods should be evaluated, adapted, and applied to future research projects that explore the human impact of significant events.

## Figures and Tables

**Figure 1 ijerph-21-00636-f001:**
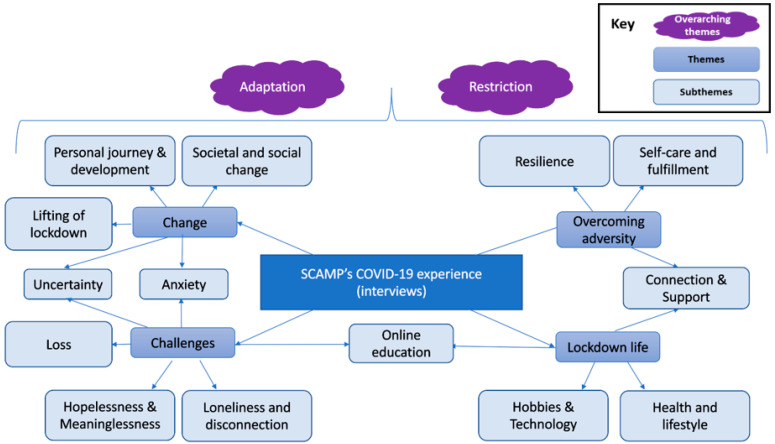
Thematic map of participant experiences of COVID-19 expressed verbally through interviews.

**Figure 2 ijerph-21-00636-f002:**
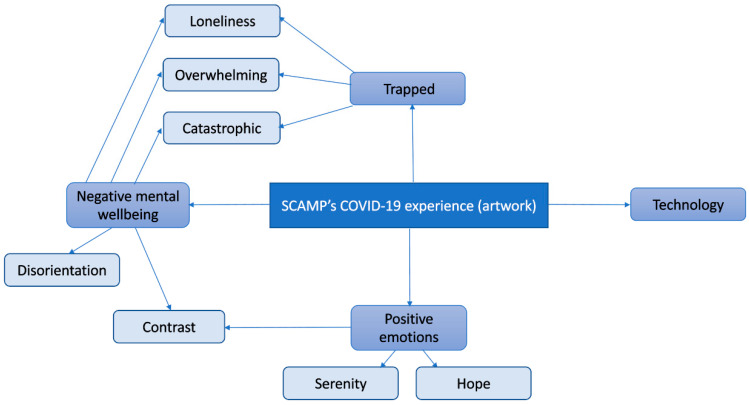
Thematic map of participant experiences of COVID-19 expressed through artwork.

**Figure 3 ijerph-21-00636-f003:**
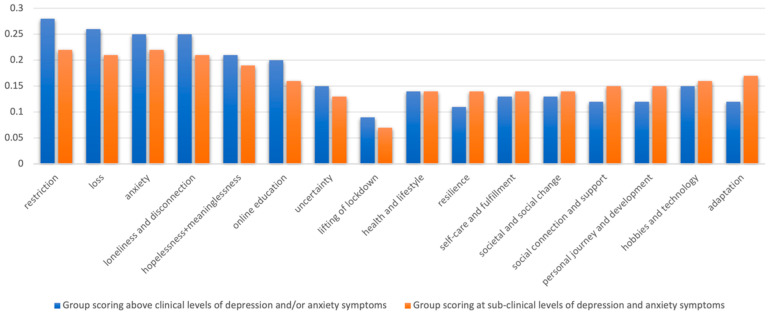
Proportions of data coded under each theme for participants with and without high depression and/or anxiety symptoms measured during the COVID-19 pandemic.

**Figure 4 ijerph-21-00636-f004:**
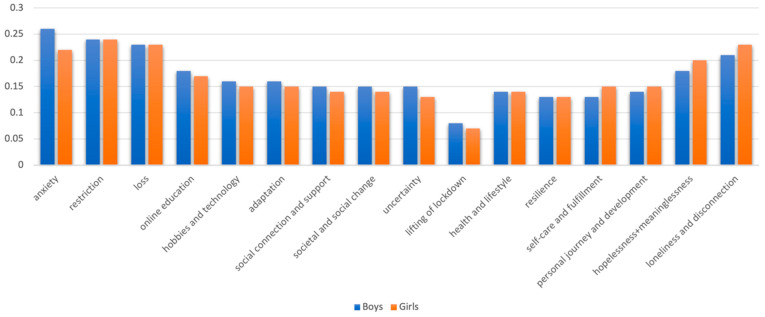
Proportion of data coded under each theme for boys and girls.

**Figure 5 ijerph-21-00636-f005:**
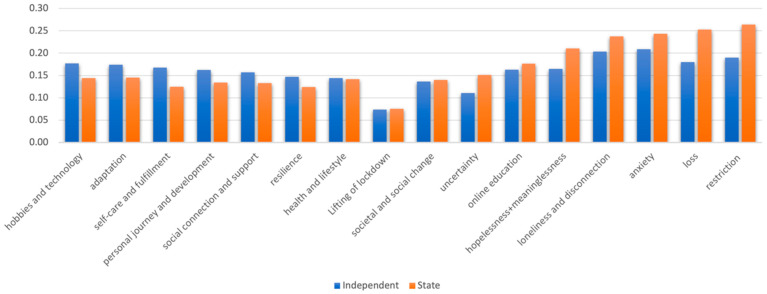
Proportion of data coded under each theme for independent and state school students.

**Table 1 ijerph-21-00636-t001:** Self-reported participant sociodemographics and health.

Characteristic	Frequency (%)
Gender	
Male	8 (38%)
Female	13 (62%)
Ethnicity	
Arab	1 (5%)
Asian Bangladeshi	5 (24%)
Asian Chinese	4 (19%)
Asian Indian	3 (14%)
Asian Pakistani	1 (5%)
Asian Other	1 (5%)
Mixed Ethnicity	1 (5%)
White British	3 (14%)
White Other	1 (5%)
Undisclosed	1 (5%)
School type	
Independent School	8 (38%)
State School	13 (62%)
Health condition or disability	
Yes	4 (19%)
No	17 (81%)
“Severe” or “moderately severe” score on the PHQ-9 (depression symptoms) and/or GAD-7 (anxiety)	
Yes	7 (33%)
No	14 (67%)
**Characteristic**	**Mean ± SD**
Age	17.73 ± 0.98
PHQ-9 depression score	10.29 ± 8.04
GAD-7 anxiety score	9.19 ± 6.49

## Data Availability

The original recordings/transcripts are not publicly available due to SCAMP’s privacy policy and ethical restrictions. The full code list from this analysis and corresponding data can be made available upon request.

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
