# Peer review of "Change, Adversity, and Adaptation: Young People’s Experience of the COVID-19 Pandemic Expressed through Artwork and Semi-Structured Interviews"

_ijerph, 2024, doi:10.3390/ijerph21050636_

Round 1
Reviewer 1 Report
Comments and Suggestions for Authors
Dear authors,
First and foremost, I would like to express my appreciation for your work. It sounds interesting and novel. However, as a reviewer for your manuscript, I aim to provide clear feedback to enhance the quality of your work. During the review process, I considered several points:
Clarity in presenting your ideas and results.
Accuracy in presenting data.
Readability of your text, especially for non-scientific community members and young researchers.
Here are my concerns:
Introduction:
Line 42 – Provide a citation for the death rate.
Line 46 – What does the term GCSEs stand for?
Lines 57 to 60, up to citation [10], seem redundant; you can merge them with lines 49-51.
Line 61 – The term “fully” does not resonate with any scientific notion; I believe that this is a strong claim to say “fully”; I would personally prefer another term rather than “fully.”
The second paragraph is too wordy; consider revising it.
Lines 66 and 77 – The term “creative methods” is broad and not defined correctly; it could be considered as limitless methods, and definitions provide a definition for it before using it.
Line 80 – Authors claim that art expression has not been combined with verbal account. I wonder if they are mentioning just this specific case or the general view? In terms of psychological experiments, as my memory helps, various studies (Vich 1967, Martinez 2013, McGlone 2012) have discussed and shared it.
Also, later, the following refs, which I mentioned earlier, suggest considering biases in this term and combining art and expressions; how would authors control these biases?
The introduction lacks a clear point to the main point/s of this study. There is a lack of cohesion in the topics explained in the introduction. Topics that could be together and later explain further studies and lead to a conclusive point to reach the aim of this study are necessary for the introduction.
After reaching line 97, I couldn’t get what is exactly the point. There are two points mentioned as a gap, but they are not connected well enough to make up a clear aim for this study.
Methods:
I did not have access to supplementary materials!
Line 110 – Provide better detail of how this sub-sample in which accordance and method has been chosen and conducted.
Since 9 participants participated in the workshop with authors LB and RT, how do you think this workshop or shared information biased the results? This raises concerns about the structure of the workshop, which is not clear to me as a reader.
In the participant section, selecting 15-30 participants includes a large variability. Based on which context specifically, statistical point have you chosen 15-30? Did the authors conduct any power analysis or any statistical calculation for recruitment?
Since you also used an online survey, what measures were taken to control for the variability of your online study?
Since qualitative analysis by AR and RM, which control has been applied to check the validity of this analysis?
Analysis sections 1, 2, and 3 are too wordy.
To me, this reads more like a story than a scientific report. I appreciate your clear points and clarifications, but some main points are not necessary to mention. For example, instead of mentioning online meetings, some other missing points are here, e.g., which control was used to check the validity of data by public contributors?
Results:
Looking at your demographic data, I see a wide range of ethnicities, which by itself causes a wide range of variability in the cohesion of results. Considering how family core and cultural core can impact this, how can authors make sure that this factor does not or did not impact their result? Did they consider this?
Table 1 characteristics are not presented correctly. It is better to present it better since we have age mean and SD, so we don’t need to have a range! Or for PHQ-9 and GAD-7
- Is the text from participants corrected or not? this means that you have chosen related parts and created new sentences, or is it directly from participants?
Authors claim in line 669 that the “issue of education…was salient.” In which statistical term or analytical term can confirm this conclusion? I could not find any quantitative or qualitative point toward this, specifically that most of the outcomes except a few are almost in the same range; how does this not claim other variables, e.g., quality of life, mental disorientation, perceived stress, and rest?
Line 684, where authors claim the “negativeness of tech…. aimlessness”: could authors provide how they have found this term and relation? Why is it not because of stressors? Or other factors? Claiming this, how confirmed by their study?
Line 719 – The term "same city" is not quite resonated in your work. Claiming this is correct we have data from different cities and the same result for each city. Do we have such data for your work?
Though your sample was diverse, the lack of enough number of participants for each sample is a bigger limit which you need to either claim or explain why you have such a small population but from different ethnics.
The structure of the paragraphs is quite long, wordy, and there are no good transitions between paragraphs. Why did not authors make sentences shorter to make it easier to follow? Although the work is interesting, the method is novel but presenting needs more clarity and restructuring.
Author Response
Thank you for this positive feedback and taking the time to provide detailed comments for improvement of this work, which we have addressed point by point below.
Introduction:
- (Originally line 42) Thank you for spotting this, it has been provided.
- (Originally Line 46) We have removed the mention of specific qualifications as it didn’t seem necessary and doesn’t apply to all pupils in the UK.
- (Original lines 57-60) Thank you, we have merged with the preceding section (revised manuscript lines 53-54).
- (Original line 61) Thank you, we agree and have replaced this summary with the following: “The range of estimates, differences between populations and subgroups, and mixed experiences in these reports suggest that the psychological impact of the COVID-19 pandemic on young people may be nuanced and complex” (revised lines 62-65)
- (The second paragraph) Thank you, we have revised this paragraph accordingly (revised lines 52-69)
- (Original lines 66 and 77). Thank you, we have clarified; “Some research has evaluated young people’s participation in artistic activities as a means of understanding pandemic experiences and/or as an intervention” (lines 70-71), “research suggests artistic and creative activities have potential for helping young people to express and process pandemic-related distress, and for improving our understanding of it” (line 81-83).
- (Original Line 80). Thank you, we agree and have now clarified that we are talking about COVID-19 research, not all research (lines 84-85). Thank you for providing these references. Unfortunately, we have not been able to identify these papers from the information provided but we have discussed the potential biases in the following sections: lines 213-215, 770-776.
- Thank you for this feedback about the lack of clarity in the introduction. We have reconfigured the final couple of paragraphs of the introduction to more clearly explain how the various gaps in the literature connect to the aims and methods of this study (lines 84-105).
Methods
- (Original line 110) Thank you, this information is found in the ‘participants’ section but we have now specified to see this section for information about the subsample (line 119).
- (Original line 123) Thank you, we have provided a bit more information to clarify the structure of the workshop. We discuss potential biases in the discussion page 24 line 779-778.
- (Original Line 158-159) 15-30 was selected based on the typical required sample size for data saturation in thematic analysis, assuming some attrition. We have now explained this in 166-168.
- (Online study) The present study was nested within the COVID-19 survey study. Survey respondents were invited to partake in the present study, survey data was used for group comparisons, and the survey results (published in full elsewhere) are discussed in relation to the results of this study. We have clarified the role of the survey in text and provided the citation to published work outlining the COVID-19 survey project in more detail (lines 98-100, 159-171).
- (Analysis section) Thank you, we have revised the analysis section to be less wordy, removing some arbitrary points and clarifying other points, such as how validity was controlled (lines 207-217).
Results
- Thank you. The cohort, being representative of greater London, is extremely diverse. We have now acknowledged this potential limitation in the discussion (768-770).
- Thank you, we have edited Table 1 accordingly.
- We have now included an explanation of the quotations in lines 268-269; ‘The italicized text provides direct quotations, with an ellipsis within sections of text indi-cating that interim text has been removed for brevity.’
Discussion
- (Originally line 669) We stated it was salient in order to indicate how present the theme of online education was and the high level of distress participants reported in their discussion of it. ‘Salient’ was not the right choice of words here and we have edited this sentence to better reflect our point (revised line 800-802).
- (Original Line 684) Thank you for this. We have edited this point in the discussion to clarify what was meant here (707-711).
- (Original Line 719) Apologies for the lack of clarity here- we only have data for one city (London) and were noting that even for 16–18-year-olds in London, experiences were diverse. We have deleted this point for clarity here (718-719).
- Thank you. We believe the diversity of participants is a strength in the sense that it is representative of London and the SCAMP cohort, but we have now acknowledged how this might introduce a limitation in terms of coherence of results (Lines 769-771).
We have revised the discussion to improve the length of sentences, paragraphs, and transitions between paragraphs (revisions highlighted in yellow).
Reviewer 2 Report
Comments and Suggestions for Authors
Thank you for the opportunity to review this study entitled “Change, adversity, and adaptation: young people’s experience of the COVID-19 pandemic expressed through artwork and semi-structured interviews” (ijerph-2966085).
The paper focused on the mental health effects of the COVID-19 pandemic. Specifically, this topic was investigated by using artwork and semi-structured interviews with a sample of 21 adolescents. They were predominantly girls (62%) and had a mean age of 18 years (range 16-18)
In my opinion, the research topic is relevant, and the study is interesting. The use of artwork and semi-structured interviews is an interesting strong point of the manuscript. Parallelly, some issues need to be addressed before the paper will be suitable for publication.
· Abstract: Please, add information about the sample (Percentage of boys and girls) to provide a clear picture of what will be presented in the paper.
· Introduction: In my opinion, it would be good to refer to trend or longitudinal studies, if any. Since the authors frame this study considering the impact that COVID-19 has on a psychological level on people, I suggest some research to propose a comprehensive framework in the introduction, which should be supplemented with further literature search by the authors:
- Hyland et al., 2021; doi: 10.1016/j.psychres.2021.113905.
- Gori & Topino, 2021; doi: 10.3390/ijerph18115651
· Introduction: the general aim of this research should be clearly outlined and defined in the final part of the introduction.
· The method and results are well articulated and complete. Personally, I enjoyed these sections.
· The "discussions" section should be enriched with a greater number of bibliographic references.
Best wishes
Author Response
Thank you for this positive feedback. We have made the requested revisions and outline them point by point below.
Abstract.
- Now included in line 28.
Introduction
- Thank you, we have included some extra references and longitudinal research (lines 53-59).
- Thank you- we have restructured and rewritten the end of the introduction to more clearly identify the aims.
Discussion
- Thank you, we have added extra references in lines 714, 748, 771,780, and 813.
Reviewer 3 Report
Comments and Suggestions for Authors
The study seeks to explore how the COVID-19 pandemic has affected young people's mental health using artwork and semi-structured interviews. The study represents a valuable study in terms of conceptualization, planning and insights. My observations are presented below:
(1) The study title is good.
(2) The abstract is well-written. I suggest adding some interpretations and implications of the study findings for readability and comprehension.
(3) The introduction also needs minor modifications. I suggest sharpening the study rationale by highlighting the practice and policy implications. I suggest describing the research questions explicitly at the end of the introduction. Please also justify the use of the methods used for the data collection. Please go through the following articles to enrich the introduction section:
Tiwari, G. K., Rai, P. K., Dwivedi, A., Ray, B., Pandey, A., & Pandey, R. (2023). A Narrative Thematic Analysis of the perceived psychological distress and health outcomes in Indian adults during the early phase of the COVID-19 pandemic. Psychology: The Journal of the Hellenic Psychological Society, 28(1), 213–229. https://doi.org/10.12681/psy_hps.28062
Tiwari, G. K., Singh, A. K., Parihar, P., Pandey, R., Sharma, D. N., & Rai, P. K. (2023). Understanding the perceived psychological distress and health outcomes of children during COVID-19 pandemic. Educational and Developmental Psychologist, 40(1), 103–114. https://doi.org/10.1080/20590776.2021.1899749
(4) The methods section also needs minor changes. Please describe the methods of ascertaining validity and integrity in the research. How over-involvement in the data was resolved.
(5) The results section also needs minor improvements. Please add a table of theme-wise major study codes and their frequency.
(6) The discussion section also needs minor improvements. Authors should highlight new findings. The conclusions should focus on the reflections of the study findings. Authors are encouraged to add subtitles for study limitations, implications, and directions for future research.
(7) The references are ok.
In conclusion, the study is a good work and its findings can contribute to the literature. However, small changes are recommended.
Comments on the Quality of English LanguageMinor language editing is suggested
Author Response
Thank you for this feedback- we have addressed your requested revisions point by point below.
- Thank you.
- We agree and have added more detail about interpretations and implication to the abstract (lines 35-40).
- Thank you. We have substantially edited the introduction to improve the rationale behind the study, highlight the importance of this research in terms of implications, and more clearly justify the methods used and the overall aims. These interesting articles regard a different age group so we didn’t see an obvious place for them in the introduction, but we have incorporated the second into the discussion (line 714).
- Thank you- we have edited the ‘Analysis’ section to clarify how validity and integrity were assured and the specific triangulated roles of each researcher (207-240).
- We chose not to include a table of codes and their frequency because of the volume of data and codes; there were 177 initial codes across coders in the interview analysis and 212 codes in the artwork analysis. Please advise if we should include a table with some example codes for each theme to illustrate where they have come from.
- Thank you, we have edited the discussion substantially to make clearer the novel findings and what they reflect. We also have added subtitles.
Reviewer 4 Report
Comments and Suggestions for Authors
This is a much needed article to identity needs & experiences of teenagers who have experienced COVID from around the world.
· UK may have different laws regarding Informed consent for research - please confirm that consent was not needed from parents/guardians & assent from participants. Or is it the case that consent was solely needed from teenagers due to their sufficient understanding of research study and age.
· Line 24 (abstract) – the prior sentence describes purpose of the study but it is unclear what “this” refers to in the following sentence and appears to feed off the purpose of the previous sentence – readers can understand it as alluding to young people’s mental health, using artwork and semi structured interviewers or both. This needs to be clarified to strengthen the purpose article.
· Figures descriptions for the bar graphs are informative but are lengthy. Could be integrated into text and a figure title.
· It is not clear when the “creative workshops” occurred in relation to the summer work experience, questionnaires from Qualtrics, and interviews. It is also not clear of the “recruitment” stream between each of those Timepoints are needed to assist in clarifying and if there were any reasons for not fully completion of study (reason for drop out) if collected.
· Clarification is needed if there were any adverse events in this study.
· Line 635 – how did authors quantify “better mental health”
· Lines 749-766 appears to be content not directly related to the stated purpose of this article.
o Additionally new data is introduced here (Line 754-758) that seems more appropriate for a separate article.
o Line 761-765 may be more appropriate to add in a different article
· Overall the discussion section needs to be significantly refined to reflect the original stated purpose of the article and if needed subsections need to be added to clarify the overall discussion of the article (limitations, future direction, etc)
· Line 788-821 introduces the topic of policy and practice which appears to conflict with the title of the article (analyzing the young’s person’s experience of COVID-19). This text may be best appropriate for a different article.
Comments on the Quality of English LanguageMinor English edits - at times excessive wordiness, can be trimmed down
Author Response
|
1) SCAMP participants have provided their own informed consent from age 16, with parents providing it prior. This was approved by an ethics committee (line 121-122) and we have clarified this in lines 114-116. 2) (Abstract original line 24) Thank you, we have made this clarification (line 24). 3) We appreciate this feedback and have edited accordingly. 4) Thank you, we have revised to the methods to make clear when each stage occurred, and reasons for drop-out (Lines 171, 179, 218, 229, 241, 258-259) 5) We have now included the information that no adverse events occurred (line 259-260). 6) (original line 635) Thank you, we have clarified that this refers to depression/anxiety scores (line 651-652). 7) (original lines 749-766) Apologies, the original submitted manuscript ends at line 717 and although the version formatted by the journal ends at 850, the lines 749-766, 754-758, and 761-765 don’t appear to correspond to the feedback provided. We have substantially edited the discussion for brevity and clarity so hopefully this has addressed the concerns raised here. 8) Thank you. We have revised the discussion substantially to clarify the purpose of each section in relation to the overall article and added subsections. 9) (Original lines 788-821) Thank you. We have clarified in the introduction that one important reason improving our understanding of young people’s experiences is to better support people in future (lines 65-69). We think the public health implications of this work are extremely important to draw attention to and include, especially in the context of IJERPHs scope and for its audience, so have retained the public and policy implications section. However, we hope substantial revisions to the introduction and discussion have made clearer how this fits into the broader aims and intended impact of this work. 10) Thank you. We have made substantial in-text revisions to address sentence length, paragraph length, and flow. |
Round 2
Reviewer 1 Report
Comments and Suggestions for Authors
Dear Authors,
Thank you so much. I am satisfied with your answers. I found your work ready to be published.